

# Comment on "Technical note: An assessment of the relative contribution of the Soret effect to open water evaporation" by Roderick and Shakespeare (2025)

Andrew S. Kowalski[1,2]

[1]Departamento de Física Aplicada Universidad de Granada, 18071, Granada, Spain
[2]Instituto Interuniversitario de Investigación del Sistema Tierra en Andalucía (IISTA), 18071, Granada, Spain

*Correspondence to*: Andrew S. Kowalski (andyk@ugr.es)

**Abstract.** This comment addresses the definition of Fick's 1st law employed in the paper "Technical note: An assessment of
the relative contribution of the Soret effect to open water evaporation" by Roderick and Shakespeare (2025), and defended by
the authors during the on-line discussion phase of their manuscript's peer review process. Based on precedence in chemical
engineering literature, the authors argue "the complete equivalence of mass- and molar-based frameworks for describing
diffusion". On the contrary, here a very simple example shows that the authors' preferred molar-based framework neglects the
key role of inertia in momentum conservation, violates Newton's laws of motion, and leads to different conclusions with regard
to isotopic discrimination. It therefore ought not be considered equivalent to the inertial framework that is consistent with the
laws of physics.

## 1 Conflicting Definitions of Fick's 1st law

Molecular diffusion is a basic concept in modern science, and so most engineers and scientists learn about Fick's 1st law early
in their education. The core idea is that in a nonuniform mixture like air, individual substances migrate down their gradients—
from high to low concentration—as a consequence of random molecular movements (Brownian motion). Importantly, such
transport is independent of any overall air movement, or wind. Thus, the increase in air humidity near an evaporating surface
causes water vapour to mix into drier air, transported principally upward over a horizontal, open-water surface. If the principle
seems straightforward, it should be noted that students in different scientific and engineering disciplines are taught at least four
conflicting versions of Fick's 1st law, and some become long attached to what they first learn. Differences among these
incompatible versions primarily regard the definition of "concentration", and are not trivial in the atmosphere because of its
compressibility and variable molar mass (Kowalski et al., 2021).
All four "concentrations" are defined as fractions, and they diverge as a function of two independent choices, one affecting the
numerator and the other the denominator. The numerator is defined either as the number of molecules (molar) or their mass
(inertial), while the denominator determines sensitivity to air's compressibility—whether the measure responds to changes in





pressure and temperature—with reference to either fluid amount or the volume that it occupies. Thus, combinations of these two choices variously define the "concentration" of a component (c) as its

- molar density ($\eta_c$; mol m$^{-3}$),
- mass density ($\rho_c$; kg m$^{-3}$),
- molar fraction ($\chi_c$; mol mol$^{-1}$), or
- mass fraction ($f_c$; kg kg$^{-1}$).

For many applications, such subtle differences are of negligible importance. However, because each of these choices affects the definition of "wind"—or how net transport is decomposed into diffusive and non-diffusive mechanisms—their differences become important for directions with very low fluid velocity components, such as perpendicular to an open-water surface. Correctly defining "concentration" is therefore critical when assessing subtleties of transport that include the Stefan flow
velocity (Kowalski, 2017) and the relative contribution of the Soret effect (Roderick and Shakespeare, 2025).

Roderick and Shakespeare (2025) describe diffusion using a molar-based framework, and the authors have argued its complete equivalence with a mass-based or inertial framework. Here a very simple example is presented that distinguishes between these two definitions of "concentration" and shows how a molar-based specification of diffusion—based on gradients in $\chi_c$—violates the laws of motion (Newton, 1846) when partitioning transport into diffusive and non-diffusive mechanisms. Disparate
transport partitioning also implies that the molar- and mass-based frameworks are not equivalent with regard to isotopic fractionation, because only diffusive transport discriminates among isotopes.

## 2 A simple case of no system motion

Let us consider a fluid whose elemental composition is 99.9% oxygen (O) by mass, with trace amounts (1 g kg$^{-1}$) of argon (Ar) whose diffusive transport in the x-direction is of interest. Initially, each element has its mass equally dispersed about the fluid's
centre of mass ($x_0$, $y_0$, $z_0$). Importantly, while atoms change position as the fluid is re-ordered, we are able to distinguish between O atoms as having originated from either the left ($x < x_0$) or right ($x > x_0$) side of the fluid, and these become redistributed by Brownian motion. After mixing, both O and Ar again have equal mass on the left and right sides, and their centres of mass still coincide at the centre point ($x_0$, $y_0$, $z_0$). Newtonian analysis indicates that the average velocities of both the O and Ar, and indeed the entire fluid, are null as a consequence of random re-ordering of mass (or "quantity of matter";
Newton, 1846), with equal amounts swapped left and right. Net transport occurs of neither O nor Ar and, from a Newtonian perspective, this situation could hardly be simpler: *there is no net motion*, neither of the fluid nor of either of its elemental components.

Now let us specify the means of discerning the left/right origin of O atoms, about which Newton knew nothing and which has no bearing on his laws. These are distinguished by chemical bond types, with O atoms initially forming dioxygen ($O_2$)
molecules on the left and triatomic ozone ($O_3$) molecules on the right. Although this is clearly a case where $O_2$ and $O_3$ must diffuse in opposite directions, the key question addressed here concerns diffusion of Ar, whose molar fraction ($\chi_{Ar}$) is 801 ppm



on the left versus 1201 ppm on the right. Whereas the mass-based definition of "concentration" (above) indicates no gradient in the Ar mass fraction $\left(\frac{\partial f_{Ar}}{\partial x} = 0\right)$, a molar-based framework for describing diffusion would find Ar more "concentrated" on the right $\left(\frac{\partial \chi_{Ar}}{\partial x} > 0\right)$. If relevant, this would imply leftward Ar diffusion and require counteracting, non-diffusive Ar transport

to the right in order to correctly describe net Ar transport, which must be null as prescribed. In short, the molar-based framework asserts that Ar diffuses upwind. But the implied wind violates the principle that momentum, or the "quantity of motion" (Newton, 1846), is conserved as zero when the system's centre of mass is static, since Newtonian physics defines velocity as the momentum-to-mass ratio.

To review, in this very simple example the inertial framework specifies no Ar transport, no Ar diffusion, and no fluid motion

as is consistent with Newton's laws. By contrast, the molar framework purports null Ar transport to be composed of leftward Ar diffusion that is offset by non-diffusive Ar transport due to a rightward fluid velocity that violates Newton's laws. The two frameworks are not equivalently consistent with the laws of physics.

The inertial and molar frameworks also disagree regarding fractionation of the various isotopes of Ar. The molar framework implies leftward Ar diffusion against a purported airflow, which would initially favour leftward transport of Ar atoms with

fewer neutrons, shifting heavier isotopes to the right. By contrast, the inertial, Newtonian framework specifies no Ar diffusion and therefore no fractionation.

In the derivations of Roderick and Shakespeare (2025), the degree to which they are not in compliance with physical laws is masked somewhat by lax adherence to Fourier's principal of dimensional homogeneity. For example, they begin their derivations introducing evaporation ($E$) and what they term its advective component ($E_A$) as *mass* fluxes in their Eq. (B1).

However, $E$ is defined in their Table 1 as a *molar* flux (with SI units that correspond to a molar flux density), and it is used as such in other equations (8, B5, B8). Particularly, $E_A$ is used to define the reference velocity in Eq. (B5), as the ratio of a molar exchanges to a molar density, and this leads to their assessment that the fraction of total water vapour transport that is not diffusive is equal to the *molar* fraction of water vapour, in their Eq. (B8). However, such a reference velocity—defined on a molar basis—overlooks the definition of the "quantity of motion" (Newton, 1846) that underlies conservation of momentum.

By contrast, Kowalski (2017) determined that the non-diffusive fraction of water vapor transport is water vapour's *mass* fraction, or specific humidity, based on Eq. (1) of that article, which is an expression of momentum conservation. Since water vapour is a light molecule within air, its molar fraction generally exceeds its mass fraction, and these conclusions regarding how diffusive and non-diffusive mechanisms contribute to net transport are not equivalent.

## 3 Conclusions

The above simple example of a case with no system motion illustrates that the molar-based framework for describing diffusion violates Newton's laws. During the on-line discussion, the authors expressed surprise at the "ferocity" of a review that recommended rejection. Perhaps such a quality is warranted in defence of the laws of physics.



**Competing interests**

The author declares that he has no competing interests.

**Acknowledgements**

The author is supported by the *Ministerio de Ciencia e Innovación* project REMEDIO (grant no. PID2021-128463OB-I00).

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
