# Peer review of "Comment on "Technical note: An assessment of the relative contribution of the Soret effect to open water evaporation" by Roderick and Shakespeare (2025)"

_EGUsphere, 2025_

## Community Comment (CC1)

**HESS Manuscript #https://egusphere.copernicus.org/preprints/2025/egusphere-2025-2814**

Title: Comment on "Technical note: An assessment of the relative contribution of the Soret effect to open water evaporation" by Roderick and Shakespeare (2025)

Author: Kowalski

**Review**

This manuscript (comment/reply) is a scientific comment on a recent paper for which I was the first author. The manuscript continues comments made by Dr Kowalski during the review of our 2025 paper and the commentary about the review process of our earlier work is accurately described in the current manuscript.

I think it might be helpful to reformulate the introductory remarks about the comment to make a new reader aware of the full context of the interesting topic raised. I attempt that below.

In their derivation, Roderick and Shakespeare (2025) used molar units and reported that the advective component of the total flux scaled with the mol fraction of water vapour near the evaporating surface. Kowalski (this manuscript and previously) has asserted that the same advective flux scales with the mass fraction of water.

To give a numerical example, assume standard air with water vapour mol fraction equal to 0.02 (i.e., reasonably warm moist air). The equivalent mass fraction of water vapour would be 0.012. The derivation of Roderick & Shakespeare (2025) would make the advective component of the flux equal to 2% of the total (molar) flux. In contrast, the analysis by Kowalski (here and previously) would make the advective component equal to 1.2% of the total (mass) flux. Hence there is a small 0.8% difference between the formulations in the chosen example. We further note that the mass fraction result advocated by Kowalski would always be smaller (by $\sim 18/29$) than the molar based result we used in our 2025 paper.

With this in mind it is important for the author to acknowledge that this small difference makes no practical difference to the conclusions made by Roderick and Shakespeare (2025) in that the Soret component of the total flux is very small as is the advective component. Using the smaller mass fraction result would actually strengthen the original conclusion. This needs to be spelled out clearly in the introduction to avoid confusion by interested readers and to establish the overall context of the comment/reply.

Instead, what the new Kowalski comment/reply points out is an interesting scientific/theoretical discrepancy that he has been active on for many years.

We hope the Kowalski comment invokes further work on this important topic.

**Reference Cited**

Roderick, M.L. and Shakespeare, C.J., 2025. Technical note: An assessment of the relative contribution of the Soret effect to open-water evaporation. Hydrol. Earth Syst. Sci., 29(8): 2097-2108.

**Michael L. Roderick, 15/7/2025**

---

## Author Comment (AC1)

Reply to comment by Roderick regarding HESS Manuscript
https://egusphere.copernicus.org/preprints/2025/egusphere-2025-2814

A. S. Kowalski

I agree with the comments made by Dr. Roderick, but only with regard to the narrow range of state conditions within which they are valid, even if those accurately describe the majority of Earth's evaporating, open-water surfaces.

Dr. Roderick's numerical example, where water vapour reaches a 2% molar fraction ($\chi_v$), does indeed represent "reasonably warm moist air", but is quite dry relative to tropical surface conditions. Extremely sultry environments reach $\chi_v = 5\%$ (Raymond et al., 2020), where practical differences between our calculations begin to emerge. Also, the Earth's hydrological system includes open-water surfaces whose temperatures exceed 60ºC, sometimes to 70ºC (Jaworowski et al., 2013), and whose extreme evaporation rates may be of interest. Air humidities in the non-turbulent layer above such surfaces far exceed that of Dr. Roderick's example. There, the fraction of water vapour transport that is not diffusive is significant, as is the difference in its estimation based on molar- versus mass-fractions (see Table 1 and Figure 1 below). Finally, the factor (~18/29) cited by Dr. Roderick increases with increasing humidity, since 29 g mol$^{-1}$ is not an accurate molecular mass for very humid air. This factor reaches unity at the boiling point, where air is water vapour.

| Variable | $p$ | $T$ | $T$ | $e_s$ | $\chi_v$ | $\rho_v$ | $p_d$ | $\rho_d$ | $\rho$ | $q$ |
|---|---|---|---|---|---|---|---|---|---|---|
| Units | mb | K | ºC | hPa | % | g m$^{-3}$ | hPa | g m$^{-3}$ | g m$^{-3}$ | % |
| | 1014.2 | 273.16 | 0.01 | 6.12 | **0.60** | 4.85 | 1008.08 | 1285.65 | 1290.50 | **0.38** |
| | 1014.2 | 283.15 | 10.00 | 12.28 | **1.21** | 9.40 | 1001.92 | 1232.70 | 1242.10 | **0.76** |
| | 1014.2 | 293.15 | 20.00 | 23.39 | **2.31** | 17.29 | 990.81 | 1177.45 | 1194.74 | **1.45** |
| | 1014.2 | 303.15 | 30.00 | 42.47 | **4.19** | 30.36 | 971.73 | 1116.68 | 1147.04 | **2.65** |
| | 1014.2 | 313.15 | 40.00 | 73.85 | **7.28** | 51.10 | 940.35 | 1046.12 | 1097.21 | **4.66** |
| | 1014.2 | 323.15 | 50.00 | 123.52 | **12.18** | 82.82 | 890.68 | 960.20 | 1043.02 | **7.94** |
| | 1014.2 | 333.15 | 60.00 | 199.46 | **19.67** | 129.73 | 814.74 | 851.96 | 981.69 | **13.21** |
| | 1014.2 | 343.15 | 70.00 | 312.01 | **30.76** | 197.01 | 702.19 | 712.87 | 909.89 | **21.65** |
| | 1014.2 | 353.15 | 80.00 | 474.14 | **46.75** | 290.91 | 540.06 | 532.75 | 823.66 | **35.32** |
| | 1014.2 | 363.15 | 90.00 | 701.82 | **69.20** | 418.74 | 312.38 | 299.67 | 718.41 | **58.29** |
| | 1014.2 | 373.15 | 100.00 | 1014.20 | **100.00** | 588.91 | 0.00 | 0.00 | 588.91 | **100.00** |

**Table 1. Comparison of water vapour's molar ($\chi_v$) and mass ($q$) fractions over an extreme humidity range. Pressure ($p$) is specified as 1014.2 mb, as are temperatures ($T$). Saturation vapour pressure ($e_s$) is from Haynes et al. (2014); for simplicity, 100% relative humidity is assumed. The ratio $e_s/p$ determines $\chi_v$. The ideal gas law calculates $\rho_v$ as $e_s/(R_v \cdot T)$ where $R_v = 461.52$ J kg$^{-1}$ K$^{-1}$ and also $\rho_d$ as $p_d/(R_d \cdot T)$ where $R_d = 287.05$ J kg$^{-1}$ K$^{-1}$, and $p_d$ is the partial pressure of dry air from Dalton's law ($p - e_s$). The air density ($\rho$) is the sum of those of water vapour and dry air ($\rho_v + \rho_d$). Finally, the specific humidity ($q$) is the ratio of $\rho_v$ to $\rho$. (Note that hydrothermal features in Yellowstone National Park have lower surface $p$, and therefore even higher fractions of water vapour than those corresponding to their temperatures in this table.)**

[Figure]

**Figure 1. Graphical comparison of water vapour's molar ($x_v$) and mass ($q$) fractions over an extreme humidity range (data from Table 1 above).**

**References**

Haynes, W.M., Lide, D. R., and Bruno, T. J., *CRC Handbook of Chemistry and Physics*, Taylor and Francis, Boca Raton, 2014.

Jaworowski, C., et al., Temporal and seasonal variations of the hot spring basin hydrothermal system, Yellowstone National Park, USA, *Remote Sens., 5*, 6587-6610, https://doi.org/10.3390/rs5126587, 2013.

Raymond, C., Matthews. T,. and Horton, R.M., The emergence of heat and humidity too severe for human tolerance, *Sci Adv* **6** eaaw1838. https://www.science.org/doi/10.1126/sciadv.aaw1838, 2020.